# LINEARISED IMPLICIT VARIATIONAL INFERENCE

## ABSTRACT

Bayesian neural networks (BNNs) are touted for robustness under data drift, resilience to overfitting and catastrophic forgetting whilst also producing actionable uncertainty estimates. In variational inference, these elegant properties are contingent on the expressivity of the variational approximation. Posteriors over parameters of large models are usually multimodal and highly correlated and hence cannot be well-approximated by simple, prescribed densities. We posit implicit variational distributions specified using differentiable generators are more flexible and propose a novel bound for training BNNs using such approximations (amortized neural samplers). The proposed bound uses an approximation of the variational distribution's entropy by locally linearising the generator. Unlike existing works, our method does not require a discriminator network and moves away from an unfavourable adversarial objective. Our formulation resembles normalizing flows but does not necessitate invertibility of the generator. Moreover, we use a differentiable numerical lower bound on the Jacobians of the generator, mitigating computational concerns. We report log-likelihoods on UCI datasets competitive with deep ensembles and test our method on out-of-distribution benchmarks.

## 1 INTRODUCTION

Deep neural networks are considered state of the art in numerous tasks in computer vision, speech and natural language processing. Scaling up neural architectures has led to outstanding performance on a myriad of generative and discriminative tasks, albeit some fundamental flaws remain. Neural networks are usually trained by maximising likelihood resulting in a single best estimate of parameters which renders these models highly overconfident of their predictions, prone to adversarial attacks and unusable in risk-averse domains. Furthermore, their usage remains restricted in sequential learning applications due to *catastrophic forgetting* (McCloskey & Cohen, 1989) and data-scarce regimes due to overfitting. When deployed in the wild, deep networks do not output a comprehensive measure of their uncertainty, prompting expert intervention.

The Bayesian paradigm provides solutions to a number of these issues. In summary, Bayesian neural networks specify a prior distribution over parameters $p(\theta)$, and the neural network relates the parameters to the data $D$ through a likelihood $p(D|\theta)$. The goal is to infer a conditional density over the parameters, called the posterior $p(\theta|D)$, given by the Bayes' rule,

$$p(\theta|\mathcal{D}) = \frac{p(\mathcal{D}|\theta)p(\theta)}{p(\mathcal{D})} = \frac{p(\mathcal{D}|\theta)p(\theta)}{\int p(\mathcal{D}, \theta)\, \mathrm{d}\theta}. \tag{1}$$

This conditional density provides a range of suitable parameters with a probability over them given by the dataset. After training, predictions from an ensemble of parameters (models) can then be combined, weighted by their posterior probability forming a Bayesian model average (BMA). The variance of these aggregated predictions informs the user/human about the model's confidence in a particular prediction. Finding the normalization constant in eq. (1) is analytically intractable for large models, and hence there is a clear focus on approximate inference techniques. Various approaches have been proposed, including Markov chain Monte Carlo (MCMC, Neal, 1995), variational inference (VI, Saul et al., 1996; Peterson, 1987) and the Laplace approximation (Mackay, 1991).

Variational inference is a strategy that converts the inference problem into an optimisation over a family of distributions (variational family), denoted hereafter by $\mathcal{Q}$, indexed by variational parameters denoted by $\gamma$. We optimise $\gamma$ using a lower bound on the marginal log-likelihood of the data $\log p(\mathcal{D})$ called the evidence lower bound (ELBO). Usually, we are computationally limited to choosing simple

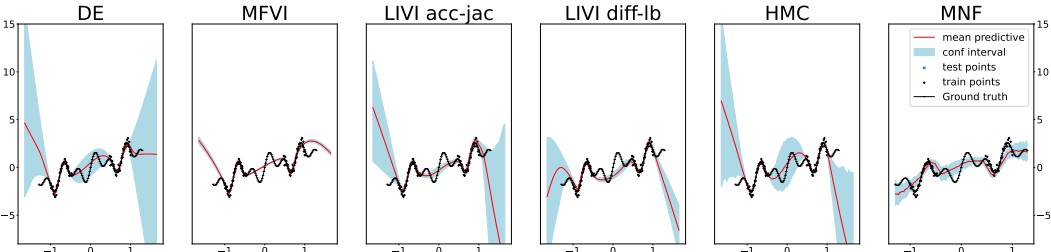

Figure 1: **Model confidence on toy regression.** We compare both lower bounds for implicit variational inference presented (LIVI) in this work with standard uncertainty quantification approaches deep ensembles (DE), mean-field variational inference (MVI), Hamiltonian Monte Carlo (HMC) and multiplicative normalizing flows (MNF, Louizos & Welling, 2017). We train using 25 dimensional noise input to a MLP generator having a single hidden layer consisting of 25 weights modeling 105 neural network parameters. Here we plot the 5-95% percentile of the BMA to represent model uncertainty. Notably both our approaches capture in-between uncertainties (Foong et al., 2019).

distribution families like an isotropic Gaussian distribution (Tanaka, 1998; Blundell et al., 2015). The true posterior is much more complex and is approximated poorly using such approximations (Foong et al., 2019; 2020). This issue is exacerbated in large models that contain many symmetries and correlations. Notably, there have been attempts to extend VI to more structured and expressive distributions (Saul & Jordan, 1995; Bishop et al., 1997; Louizos & Welling, 2016) yet, capturing correlations between parameters with a flexible variational approximation remains the Achilles heel of these class of models.

We propose an approach based on implicit generative modelling where the distribution over variables of interest is implicit and can only be sampled. This is in contrast to usual VI methods that use prescribed distributions with explicit parametrisation as the approximating density over latent variables (Diggle & Gratton, 1984; Mohamed & Lakshminarayanan, 2016). Although, this idea takes inspiration from GAN generators that try to recover the true data distribution, we do not require a discriminator network for training the generator and as a result do not suffer from the complicacies introduced by an adversarial objective. As emphasised by Tran et al. (2017), is a more natural way of capturing the generative process instead of forcing it to conform to an assumed latent structure which could be misspecified.

Similar to other works in implicit VI (Shi et al., 2018), we posit using general (non-invertible) stochastic transformations that can produce highly flexible implicit densities to model posteriors of neural networks. We believe that these approximations can better capture the intricacies of posterior landscape. Additionally, when trying to model complicated densities in high-dimensions, it is sensible to learn a sampler instead of parameters of an expressive intractable approximation, especially if these approximations do not admit *one-line samplers* (Devroye, 1996). For example, EBMs can be very flexible but are not easy to sample from (Song & Kingma, 2021).

If we were to use a fully correlated Gaussian to model the posterior of a neural network, we would need to optimize parameters quadratic in the number of weights of the network, $O(\dim(\theta)^2)$ to arrive at the optimum covariance matrix. In this work, we test our hypothesis of using an under-parameterised generator to capture the important correlations in orders of magnitude less parameters than that. At the same time, we hint at the possibility that a constrained generator will probably avoid modelling redundancies present in BNN posteriors like permutationally symmetric modes.

Succinctly, our contributions are presented as follows:

- We derive a novel lower bound for variational inference in Bayesian Neural Networks using implicit variational approximation avoiding unstable minmax objectives.
- We augment this lower bound by reducing its compute requirement, as we substitute a differentiable numerical lower bound for the entropy term comprising of Jacobians of neural networks.
- We comprehensively empirically evaluate the capacity of this implicit variational approximation and the quality of the posteriors inferred using different out of distribution benchmarks.

## 2 VARIATIONAL INFERENCE FOR BAYESIAN NEURAL NETWORKS

Consider the supervised learning setting, where we have training set $\mathcal{D} = \{(\boldsymbol{x}_i, \boldsymbol{y}_i)\}_{i=1}^n$, where $\boldsymbol{X} = \{\boldsymbol{x}_i\}_{i=1}^n$ are the covariates (inputs) and $\boldsymbol{y} = \{y_i\}_{i=1}^n$ are the labels (outputs). We consider a Bayesian regression or classification model given by

$$p(\mathcal{D}, \boldsymbol{\theta}) = p(\boldsymbol{\theta})p(\mathcal{D}|\boldsymbol{\theta}) = p(\boldsymbol{\theta}) \prod_{i=1}^n p(y_i|\boldsymbol{x}_i, \boldsymbol{\theta}), \tag{2}$$

where the likelihood is parameterised by $\boldsymbol{\theta} \in \Theta \equiv \mathbb{R}^m$. The objective function $\mathcal{L}$ in VI, called the ELBO is a lower bound on the log marginal likelihood of the data - $\log p(\mathcal{D})$, and the discrepancy between the two is equal to the KL divergence between the approximate and true posterior given by

$$D_{\mathrm{KL}}[q_\gamma(\boldsymbol{\theta})||p(\boldsymbol{\theta}|\mathcal{D})] = \log p(\mathcal{D}) - \underbrace{\mathbb{E}_{\boldsymbol{\theta} \sim q_\gamma(\gamma)} \left[ \log \frac{p(\mathcal{D}, \boldsymbol{\theta})}{q_\gamma(\boldsymbol{\theta})} \right]}_{\mathcal{L}(\gamma)}, \tag{3}$$

where $q_\gamma$ is the variational approximation of the posterior with parameters $\gamma \in \Gamma$. Since the KL divergence is non-negative, $\mathcal{L}$ is a lower bound on the evidence. This objective function can be written in terms of a *likelihood term* and a *regularisation term* as

$$\mathcal{L}(\gamma) = \left[ \underbrace{\mathbb{E}_{\boldsymbol{\theta} \sim q_\gamma(\boldsymbol{\theta})} \left[ \log p(\mathcal{D}|\boldsymbol{\theta}) \right]}_{\text{likelihood term}} - \underbrace{D_{\mathrm{KL}}[q_\gamma(\boldsymbol{\theta})||p(\boldsymbol{\theta})]}_{\text{regularisation term}} \right] \leq \log p(\mathcal{D}), \tag{4}$$

where the *likelihood term* promotes the variational approximation to model the data well and the *regularisation term* keeps the posterior close to the prior. Since the log-evidence, $\log p(\mathcal{D})$, does not depend on $\gamma$, minimising the the KL divergence is equivalent to maximising the ELBO, i.e.,

$$\arg\min_\gamma D_{\mathrm{KL}}[q_\gamma(\boldsymbol{\theta})||p(\boldsymbol{\theta}|\mathcal{D})] \equiv \arg\max_\gamma \mathcal{L}(\gamma). \tag{5}$$

### 2.1 IMPLICIT VARIATIONAL INFERENCE

In implicit VI (IVI), the variational distribution is only implicitly defined through its generative process over the parameters $\boldsymbol{\theta}$

$$\boldsymbol{z} \sim q(\boldsymbol{z}), \quad \boldsymbol{\theta} = g_\gamma(\boldsymbol{z}), \tag{6}$$

where the $q$ is a fixed base distribution and $g_\gamma : \mathbb{R}^d \to \mathbb{R}^m$ is a non-linear mapping and typically not a diffeomorphism. For IVI, the *likelihood term* from eq. (4) and its gradients can be estimated using Monte Carlo and the reparameterization trick. However, the *regularisation term* is more difficult as it involves the entropy of $q_\gamma$,

$$D_{\mathrm{KL}}[q_\gamma(\boldsymbol{\theta})||p(\boldsymbol{\theta})] = \mathbb{E}_{\boldsymbol{\theta} \sim q_\gamma(\boldsymbol{\theta})} \left[ \log \frac{q_\gamma(\boldsymbol{\theta})}{p(\boldsymbol{\theta})} \right] = -H_q(q_\gamma) - \mathbb{E}_{\boldsymbol{\theta} \sim q_\gamma(\boldsymbol{\theta})} \left[ \log p(\boldsymbol{\theta}) \right]. \tag{7}$$

Generally, the entropy of the generative process in eq. (6) is not available in an explicit form as the density of the process is not tractable. A prevalent technique to estimate the *regularisation term* uses density ratio estimators based on a GAN-like discriminator(Sugiyama et al., 2012; Huszár, 2017), and Geng et al. (2021) have given a tractable and differentiable lower bound on this entropy .

Furthermore, when the dimensions of the base distribution $d$ is smaller than $m$, the KL divergence is not well defined. In the KL divergance, we integrate over the whole space $\Theta$ but $q_\gamma$ does not have full support over this space and exists on a manifold embedded in the $\Theta$ space. In the GAN literature this problem is called *mode dropping* and is caused by the inability of the generator to recover all modes of the true data distribution (Che et al., 2020; Xu et al., 2018). To alleviate this, we draw inspiration from works in the GAN literature (Che et al., 2020) and add $m$ dimensional noise to the output of the generator and redefine the variational approximation in the following section.

## 3 A DEEP LATENT VARIABLE MODEL AND ITS ENTROPY

As the variational distribution, we propose to use a Gaussian deep latent variable model (DLVM) of a real variable $\boldsymbol{\theta} \in \mathbb{R}^m$ and with a real latent variable $\boldsymbol{z} \in \mathbb{R}^d$ with density

$$q(\boldsymbol{\theta}) = \int q(\boldsymbol{\theta}|\boldsymbol{z})q(\boldsymbol{z})\,\mathrm{d}\boldsymbol{z} = \mathbb{E}_{\boldsymbol{z} \sim q(\boldsymbol{z})}[q(\boldsymbol{\theta}|\boldsymbol{z})]. \tag{8}$$

We assume a Gaussian base density and a Gaussian output density, that is

$$q(\boldsymbol{z}) = \mathcal{N}(\boldsymbol{z}|\mathbf{0}, I_d) \tag{9}$$

$$q(\boldsymbol{\theta}|\boldsymbol{z}) = \mathcal{N}(\boldsymbol{\theta}|g_\gamma(\boldsymbol{z}), \sigma^2 I_m), \tag{10}$$

where $g : \mathbb{R}^d \to \mathbb{R}^m$ is the decoder/generator and $\sigma^2 \in \mathbb{R}^+$ is the fixed homoscedastic variance of the output density. In general, we do not have a closed form for $q(\boldsymbol{\theta})$ due to the the integral in eq. (8) and the non-linear $g_\gamma$, but we note that KL divergence in eq. (7) is well defined for this variational distribution. Below we propose a novel approximation of the differential entropy of this model.

This model can equivalently be viewed as a variational autoencoder (VAE, Kingma & Welling, 2014; Rezende et al., 2014) with a Gaussian prior and a Gaussian output density with constant constant homoscedastic variance and no encoder, or as a implicit distribution from eq. (6) with added Gaussian noise. The latter is clearly seen from the generative process of by describe the generative process for The generative process of $\boldsymbol{\theta}$, which is

$$\boldsymbol{\theta}' = g_\gamma(\boldsymbol{z}), \quad \boldsymbol{z} \sim \mathcal{N}(0, I_d) \tag{11}$$

$$\boldsymbol{\theta} = \boldsymbol{\theta}' + \boldsymbol{\eta}, \quad \boldsymbol{\eta} \sim \mathcal{N}(0, \sigma^2 I_m). \tag{12}$$

### 3.1 DIFFERENTIAL ENTROPY

We want to calculate the different entropy of the Gaussian DLVM given by

$$H[q(\boldsymbol{\theta})] = -\mathbb{E}_{\boldsymbol{\theta} \sim q(\boldsymbol{\theta})}[\log q(\boldsymbol{\theta})]. \tag{13}$$

We can in general not compute this analytically since we do not have a closed form of $p(\boldsymbol{\theta})$. Since we can sample from $p(\boldsymbol{\theta})$, we can approximate the expectation in eq. (13) using Monte Carlo sampling from $p(\boldsymbol{z})$. However, since we do not have a closed form of $p(\boldsymbol{\theta})$, we still need an approximation of $\log q(\boldsymbol{\theta})$. We could approximate $p(\boldsymbol{\theta})$ using Monte Carlo sampling from $p(\boldsymbol{z})$, but this approximation has high variance. Usually, the variance is reduced by learning and encoder and doing importance sampling. Here we derive an approximation without using an encoder.

#### 3.1.1 LINEARISATION OF THE GENERATOR

First we consider a local linearisation of the generator. Assuming that the Jacobian of $g$ exists, the first order Taylor polynomial of $g$ at $\boldsymbol{z}_0$ is given by

$$T_{\boldsymbol{z}_0}^1(\boldsymbol{z}) = g(\boldsymbol{z}_0) + \mathrm{D}g(\boldsymbol{z}_0)\,(\boldsymbol{z} - \boldsymbol{z}_0), \tag{14}$$

where $\mathrm{D}g(\boldsymbol{z}_0)$ is the Jacobian of $g$ evaluated in $\boldsymbol{z}_0$. This assumes that the Jacobian exists, i.e. the generator has at least one derivative. We can approximate $g(\boldsymbol{z})$ by $T_{\boldsymbol{z}_0}^1(\boldsymbol{z})$ when $\boldsymbol{z}$ is close to $\boldsymbol{z}_0$. We apply this approximation to $q(\boldsymbol{\theta})$ from eq. (8), which gives us

$$q(\boldsymbol{\theta}) = \mathbb{E}_{\boldsymbol{z} \sim q(\boldsymbol{z})}[q(\boldsymbol{\theta}|\boldsymbol{z})] = \mathbb{E}_{\boldsymbol{z} \sim q(\boldsymbol{z})}[\mathcal{N}(\boldsymbol{\theta}|g(\boldsymbol{z}), \sigma^2 I_m)] \tag{15}$$

$$\approx \mathbb{E}_{\boldsymbol{z} \sim q(\boldsymbol{z})}[\mathcal{N}(\boldsymbol{\theta}|g(\boldsymbol{z}_0) + \mathrm{D}g(\boldsymbol{z}_0)\,(\boldsymbol{z} - \boldsymbol{z}_0), \sigma^2 I_m)] = \mathcal{N}(\boldsymbol{\theta}|\mu(\boldsymbol{z}_0), C(\boldsymbol{z}_0)) =: \tilde{q}_{\boldsymbol{z}_0}(\boldsymbol{\theta}), \tag{16}$$

where

$$\mu(\boldsymbol{z}_0) = g(\boldsymbol{z}_0) - \mathrm{D}g(\boldsymbol{z}_0)\,\boldsymbol{z}_0 \tag{17}$$

$$C(\boldsymbol{z}_0) = \mathrm{D}g(\boldsymbol{z}_0)\,\mathrm{D}g(\boldsymbol{z}_0)^\intercal + \sigma^2 I_m. \tag{18}$$

The result in eq. (16) can be obtained analytically by integrating over the latent variable, see e.g. Tipping & Bishop (1999).

### 3.2 Approximation of the differential entropy

We use the Gaussian approximation of $q(\boldsymbol{\theta})$ to approximate the entropy of the DLVM, that is

$$H[q(\boldsymbol{\theta})] = -\mathbb{E}_{\boldsymbol{z}\sim q(\boldsymbol{z})}\mathbb{E}_{\boldsymbol{\theta}\sim q(\boldsymbol{\theta}|\boldsymbol{z})}[\log q(\boldsymbol{\theta})] \approx -\mathbb{E}_{\boldsymbol{z}\sim q(\boldsymbol{z})}\mathbb{E}_{\boldsymbol{\theta}\sim q(\boldsymbol{\theta}|\boldsymbol{z})}[\log \tilde{q}_{\boldsymbol{z}_0=\boldsymbol{z}}(\boldsymbol{\theta})] =: \tilde{H}[p(\boldsymbol{\theta})]. \quad (19)$$

Importantly, we do the linearisation of $q(\boldsymbol{\theta})$ around the latent value $\boldsymbol{z}$ that is used to sample each $\boldsymbol{\theta}$ in the expectation. We have that

$$\log \tilde{q}_{\boldsymbol{z}_0=\boldsymbol{z}}(\boldsymbol{\theta}) = -\frac{p}{2}\log 2\pi - \frac{1}{2}\log \det C(\boldsymbol{z}_0) - \underbrace{\frac{1}{2}(\boldsymbol{\theta} - \boldsymbol{\mu}(\boldsymbol{z}_0))^\intercal C(\boldsymbol{z}_0)^{-1}(\boldsymbol{\theta} - \boldsymbol{\mu}(\boldsymbol{z}_0))}_{=:h(\boldsymbol{\theta},\boldsymbol{z}_0)}, \quad (20)$$

which means that our approximation of the entropy is

$$\tilde{H}[q(\boldsymbol{\theta})] = \frac{m}{2}\log 2\pi + \frac{1}{2}\mathbb{E}_{\boldsymbol{z}\sim q(\boldsymbol{z})}[\log \det C(\boldsymbol{z})] + \mathbb{E}_{\boldsymbol{z}\sim q(\boldsymbol{z})}\mathbb{E}_{\boldsymbol{\theta}\sim q(\boldsymbol{\theta}|\boldsymbol{z})}[h(\boldsymbol{\theta},\boldsymbol{z})]. \quad (21)$$

As shown in appendix A.1, the last term can be written as

$$\mathbb{E}_{\boldsymbol{z}\sim q(\boldsymbol{z})}\mathbb{E}_{\boldsymbol{\theta}\sim q(\boldsymbol{\theta}|\boldsymbol{z})}[h(\boldsymbol{\theta},\boldsymbol{z})] = \frac{1}{2}\mathbb{E}_{\boldsymbol{z}\sim q(\boldsymbol{z})}\left[\mathrm{tr}\left(\left(\mathrm{D}g(\boldsymbol{z})^2 + \sigma^2 I_m\right)^{-1}\left(\sigma^2 I_m + (\mathrm{D}g(\boldsymbol{z})\,\boldsymbol{z})^2\right)\right)\right], \quad (22)$$

where we used the notation $M^2 = MM^\intercal$ for a matrix $M$. Now, if we let $\sigma^2$ tend to zero, we find that

$$\lim_{\sigma^2\to 0}\mathbb{E}_{\boldsymbol{z}\sim q(\boldsymbol{z})}\mathbb{E}_{\boldsymbol{\theta}\sim q(\boldsymbol{\theta}|\boldsymbol{z})}[g(\boldsymbol{\theta},\boldsymbol{z})] = \frac{1}{2}\mathbb{E}_{\boldsymbol{z}\sim q(\boldsymbol{z})}\left[\mathrm{tr}\left(\left(\mathrm{D}g(\boldsymbol{z})^2\right)^{-1}(\mathrm{D}g(\boldsymbol{z})\,\boldsymbol{z})^2\right)\right] \quad (23)$$

$$= \frac{1}{2}\mathbb{E}_{\boldsymbol{z}\sim q(\boldsymbol{z})}\left[\boldsymbol{z}^\intercal\boldsymbol{z}\right] = \frac{1}{2}\mathrm{tr}(I_d) = \frac{d}{2}. \quad (24)$$

Similar, we can also take the limit of the determinant term from eq. (21), that is

$$\lim_{\sigma^2\to 0}\frac{1}{2}\mathbb{E}_{\boldsymbol{z}\sim q(\boldsymbol{z})}[\log \det C(\boldsymbol{z})] = \frac{1}{2}\mathbb{E}_{\boldsymbol{z}\sim q(\boldsymbol{z})}\left[\log \det\left(\mathrm{D}g(\boldsymbol{z})\,\mathrm{D}g(\boldsymbol{z})^\intercal\right)\right] \quad (25)$$

Combining eqs. (21), (24) and (25), gives us the final approximation. For small values of the output variance $\sigma^2$, we can approximate the differential entropy of a DLVM as

$$H[p(\boldsymbol{\theta})] \approx \lim_{\sigma^2\to 0}\tilde{H}[p(\boldsymbol{\theta})] = \frac{d}{2} + \frac{m}{2}\log 2\pi + \frac{1}{2}\mathbb{E}_{\boldsymbol{z}\sim q(\boldsymbol{z})}\left[\log \det\left(\mathrm{D}g(\boldsymbol{z})\,\mathrm{D}g(\boldsymbol{z})^\intercal\right)\right]. \quad (26)$$

We can get a slightly more accurate approximation, by only applying the limit from eq. (23), and not the limit from eq. (25), which gives us

$$H[p(\boldsymbol{\theta})] \approx \frac{d}{2} + \frac{m}{2}\log 2\pi + \frac{1}{2}\mathbb{E}_{\boldsymbol{z}\sim q(\boldsymbol{z})}\left[\log \det\left(\mathrm{D}g(\boldsymbol{z})\,\mathrm{D}g(\boldsymbol{z})^\intercal + \sigma^2 I_m\right)\right]. \quad (27)$$

## 4 Linearised Implicit Variational Inference (LIVI)

We propose a novel bound for IVI. As the variational distribution, we use the DLVM of eq. (8), which is equivalent to adding noise to the implicit distribiuion of eq. (6). Using the entropy approximation from eq. (26), we propose the approximate ELBO,

$$\tilde{\mathcal{L}}(\gamma) = \mathbb{E}_{\boldsymbol{\theta}\sim q_\gamma(\boldsymbol{\theta})}\left[\log p(\mathcal{D}|\boldsymbol{\theta})\right] + \mathbb{E}_{\boldsymbol{\theta}\sim q_\gamma(\boldsymbol{\theta})}\left[\log p(\boldsymbol{\theta})\right] + \lim_{\sigma^2\to 0}\tilde{H}[p(\boldsymbol{\theta})] \quad (28)$$

$$= \mathbb{E}_{\boldsymbol{\theta}\sim q_\gamma(\boldsymbol{\theta})}\left[\log p(\mathcal{D}|\boldsymbol{\theta}) + \log p(\boldsymbol{\theta})\right] + \frac{1}{2}\mathbb{E}_{\boldsymbol{z}\sim q(\boldsymbol{z})}\left[\log \det\left(\mathrm{D}g(\boldsymbol{z})\,\mathrm{D}g(\boldsymbol{z})^\intercal\right)\right] + c, \quad (29)$$

where $c = \frac{d}{2} + \frac{m}{2}\log 2\pi$. We can reparameterise the above with the base variables $\boldsymbol{z}, \boldsymbol{\eta}$ to get

$$\tilde{\mathcal{L}}(\gamma) = \mathbb{E}_{\boldsymbol{z}\sim q(\boldsymbol{z}),\boldsymbol{\eta}\sim q(\boldsymbol{\eta})}\left[\log p(\mathcal{D}|g(\boldsymbol{z}) + \eta) + \log p(g(\boldsymbol{z}) + \eta) + \frac{1}{2}\log \det\left(\mathrm{D}g(\boldsymbol{z})\,\mathrm{D}g(\boldsymbol{z})^\intercal\right)\right] + c. \quad (30)$$

To avoid the calculation of the log-determinant term, we can follow Geng et al. (2021, eq. 10) and lower-bound it as

$$\frac{1}{2} \log \det(\mathrm{D}g(\boldsymbol{z}) \, \mathrm{D}g(\boldsymbol{z})^{\intercal}) = \frac{1}{2} \sum_{i=1}^{m} \log s_i^2(\boldsymbol{z}) \geq m \log s_1(\boldsymbol{z}), \tag{31}$$

where $s_m(\boldsymbol{z}) \geq \ldots \geq s_1(\boldsymbol{z})$ are the singular values of the Jacobian $\mathrm{D}g(\boldsymbol{z})$. This gives us a lower bound on $\tilde{\mathcal{L}}(\gamma)$ given by

$$\tilde{\tilde{\mathcal{L}}}(\gamma) = \mathbb{E}_{\boldsymbol{\theta} \sim q_\gamma(\boldsymbol{\theta})} \left[ \log p(\mathcal{D}|\boldsymbol{\theta}) + \log p(\boldsymbol{\theta}) \right] + \mathbb{E}_{\boldsymbol{z} \sim q(\boldsymbol{z})} \left[ m \log s_1(\boldsymbol{z}) \right] + c \leq \tilde{\mathcal{L}}(\gamma) \tag{32}$$

Again, by reparameterisation with $\boldsymbol{z}, \boldsymbol{\eta}$ we get

$$\tilde{\tilde{\mathcal{L}}}(\gamma) = \mathbb{E}_{\boldsymbol{z} \sim q(\boldsymbol{z}), \boldsymbol{\eta} \sim q(\boldsymbol{\eta})} \left[ \log p(\mathcal{D}|g(\boldsymbol{z}) + \boldsymbol{\eta}) + \log p(g(\boldsymbol{z}) + \boldsymbol{\eta}) + m \log s_1(\boldsymbol{z}) \right] + c. \tag{33}$$

We denote $\tilde{\mathcal{L}}(\gamma)$ the LIVI bound with accurate Jacobian and $\tilde{\tilde{\mathcal{L}}}(\gamma)$ the LIVI bound with a differentiable lower bounded on the determinant. Depending on the amount of compute available, the two bounds provide a trade-off between the accuracy of uncertainties and the resources consumed. In both cases, the entropy maximisation promotes the generator to generate diverse weight samples which is in accordance with the principle behind Bayesian model averaging and supported by the performance of deep ensembles (Lakshminarayanan et al., 2017). We present connections with existing works in the literature in the following section highlighting similarities and divergences.

## 5 RELATED WORK

The usage of a secondary network to generate parameters of a primary network first appeared in the form of *hypernetworks* (Ha et al., 2017). Our approach is probabilistic and is hence closer to Bayesian hypernetworks (Krueger et al., 2017). Compared to our approach, these models require invertibility of the generator and thereby avoid the complexities of estimating the entropy term. This corresponds to using a normalizing flow as a variational approximation. Training a normalizing flow over large parameter spaces is computionally costly due to large Jacobian matrices, typically requiring particular focus on the design of the variational approximation to curb dimensionality of the flow. In particular, Louizos & Welling (2017) use an expressive flow on multiplicative factors of weights in each layer and not on all weights jointly. Our bound uses a very similar change in volume formulation, $\log \det(\mathrm{D}g(\boldsymbol{z}) \, \mathrm{D}g(\boldsymbol{z}))$, for obtaining the log probability of samples under the variational density, but does not necessitate invertibility making it more general.

Subsequently, Shi et al. (2018); Tran et al. (2017); Pawlowski et al. (2017) have successfully demonstrated implicit variational inference in BNNs using hypernetworks. Shi et al. (2018); Tran et al. (2017) do not focus on the entropy term, but rather try to estimate the ratio of the variational approximation to the prior (regularisation-term) in a procedure called density ratio estimation (also referred to as the prior-contrastive formulation by Huszár, 2017). Tran et al. (2017) opt for training a discriminator network to maximally distinguish two distributions given only i.i.d. samples from each. This approach, though general, adds to the computational requirements and becomes more challenging in high dimensions (Sugiyama et al., 2012). To mitigate the overhead of training the discriminator for each update of the ELBO, many works limit the discriminator training to a single or few iterations. Furthermore, this approach entails an adversarial objective that are infamously unstable (Mescheder et al., 2017). Pawlowski et al. (2017) treat all the weights as independent and find that a single discriminator network is inaccurate at estimating log ratios when compared to the analytical form of *Bayes by backprop* (Blundell et al., 2015), and opt to use a kernel method that matches the analytical form more closely. Shi et al. (2018) propose a novel way of estimating the ratio of the two densities using kernel regression in the space of parameters which obviates the need for a minmax objective. An obvious difficulty with kernel ridge regression in practice is its inaccuracy to estimate high-dimensional density ratios which is similar to using discriminators. This is especially the case given a limited number of samples from both the densities as well as the RBF kernel. While the RBF kernel still takes the same high-dimensional inputs and does not involve learning massive sets of parameters, its accuracy at larger scales is still doubtful. This work also proposes *matrix multiplication neural network* (MMNN) a novel generator architecture for generating large set of parameters. Pradier et al. (2018) are also motivated by the possibility of compressing the posterior in

Table 1: **UCI regression datasets.** We report RMSE and log-likelihoods on the test set and average across three different seeds for each model to quantify the variance in the results.

| | Test RMSE ↓ | | | Test LL ↑ | | |
|---|---|---|---|---|---|---|
| Dataset | LIVI | Deep Ensembles | KIVI | LIVI | Deep Ensembles | KIVI |
| Boston | $2.461 \pm 0.131$ | $3.28 \pm 1.00$ | $2.798 \pm 0.173$ | $-2.318 \pm 0.092$ | $-2.41 \pm 0.25$ | $-2.527 \pm 0.102$ |
| Concrete | $5.150 \pm 0.169$ | $6.03 \pm 0.58$ | $4.702 \pm 0.116$ | $-3.068 \pm 0.120$ | $3.06 \pm 0.18$ | $-3.054 \pm 0.043$ |
| Energy | $0.580 \pm 0.270$ | $2.09 \pm 0.29$ | $0.467 \pm 0.015$ | $-1.170 \pm 0.110$ | $-1.38 \pm 0.22$ | $-1.298 \pm 0.005$ |
| Kin8nm | $0.081 \pm 0.001$ | $0.09 \pm 0.00$ | $0.0075 \pm 0.0010$ | $1.180 \pm 0.011$ | $1.20 \pm 0.02$ | $1.162 \pm 0.008$ |
| Naval | $0.001 \pm 0.000$ | $0.000 \pm 0.000$ | $0.00 \pm 0.00$ | $5.523 \pm 0.170$ | $5.63 \pm 0.05$ | $5.501 \pm 0.121$ |

a lower dimensional space and use an inference network with a generator. Their model differs from ours as they also consider the parameters of the generator/decoder to be stochastic. Moreover they require empirical weight samples to train which doubles the training steps. D-SIVI (Molchanov et al., 2019) and SIVI (Yin & Zhou, 2018) use Monte Carlo (MC) averaging to approximate the entropy term. Both works use the implicit formulation to only model the mixing coefficients and not all the weights of the network. Our entropy term 8 also has a similar form and can be MC approximated.

In the spirit of some recent works (Izmailov et al., 2020; Daxberger et al., 2021b;a) that alternatively choose a lower dimensional representation to preclude costly, high-dimensional inference, our work can be seen as allowing the approximate posterior in the form of the generator to choose which dimensions and parts of posterior are crucial and model them accordingly.

# 6 EXPERIMENTS

## 6.1 TOY DATA

In fig. 1, we compare inference with our method against the gold standard for posterior inference on a simple toy dataset. After training, we also plot a KDE-plot of the samples the generator outputs in appendix A.6. We infer from this plot that the generator is capable of representing non-trivial distributions as we can spot heavy tails and multiple modes.

## 6.2 UCI DATASETS

We perform experiments on UCI regression datasets with the setups by Lakshminarayanan et al. (2017) and Shi et al. (2018), using a BNN with one hidden layer MLP with 50 units on all of these datasets. We report the RMSE and log-likelihood on held out data for our method. We use generator architectures that are either equally or less powerful than Shi et al. (2018) and do not assume independence across layers, i.e. using one MLP to generate all the weights of the BNN. All of the generator architectures are one hidden layered MLP with a slightly varying number of units depending on the dataset. At this scale it is feasible to estimate uncertainties using accurate Jacobians. We require far fewer number of samples (5-10) per iteration compared to 100 used by KIVI to achieve very competitive results. We suspect they use high number of samples to curb the variance of the kernel estimator.

Our results are summarised in table 1. We train our method with a homoscedastic assumption i.e. the variance in the dataset is assumed to be constant and we train an observation noise parameter using type II maximum likelihood.

## 6.3 MNIST DATASET

Next we test our method on the MNIST dataset. While using the MMNN as the generator, we were able to achieve errors on the test set on par with KIVI for MLP with 400 and 800 hidden units. With the total number of parameters generated exceeding 400K even for 400 hidden units we chose to train the model only with the differentiable lower bound due to prohibitively high memory usage. For OOD testing we compare our method to last-layer laplace, deep ensembles and a simple MAP estimate. We intentionally choose these methods to compare against as a mean-field approximation usually does not achieve good accuracy on in-distribution data and has been shown, repeatedly, to suffer from

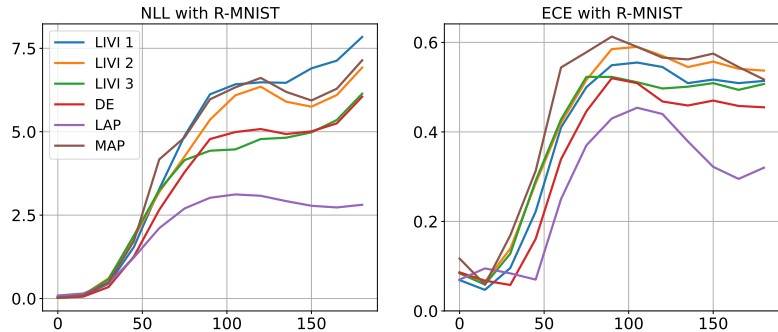

Figure 2: **Rotated MNIST benchmark.** We compare a Bayesian neural network trained via our method with MAP, deep ensembles (DE) and last-layer laplace (LAP) where our best model achieves results competitive with deep ensembles and with post-hoc method like laplace.

many optimisation difficulties. On the other hand it is possible to run HMC samplers at this scale, it is not preferable. Very few works in the literature report results using full batch HMC (Izmailov et al., 2021). Deep ensembles predict using neural networks that have converged onto different minima hence encompassing information from diverse modes of the posterior, and as such remains one of the best in terms of uncertainty estimation. As for benchmarks we choose two OOD benchmarks presented in Daxberger et al. (2021a). First we test the OOD AUROC and confidence of a LeNet5 BNN trained on MNIST by using FMNIST, KMNIST and EMNIST. We used a MMNN architecture for generating over 40K parameters and trained using the differentiable numerical approximation with 3-6 samples depending on the architecture. We expand on few generator architectures here and leave the rest for appendix appendix A.2.

The BNN trained with the implicit variational approximation, a generator with a 1225 dimension noise input and 2 matrix multiplication layers of 350 units each achieves accuracy of $99.071\% \pm 0.02$, and calibration error of $0.084 \pm 0.011$ with nll $-0.021 \pm 0.001$ on the test set. The same model reports an averaged OOD AUCROC of $97.15 \pm 0.17$ with an averaged confidence $68.53 \pm 0.24$. According to results provided in Daxberger et al. (2021a, Table 1), our model does not outperform in terms of confidence values yet, we notice that the performance degrades very smoothly as it encounters OOD data as opposed to models like Deep Ensembles and Laplace both of which fail relatively immediately and drastically in terms of confidence values. Our model does perform quite well on the averaged AUROC as well as on test set calibration error and log-likelihood.

To probe out of distribution performance further, we compare our method on the rotated MINST benchmark from Daxberger et al. (2021a). In this benchmark we plot the negative log-likelihoods and empirical calibration errors for different rotated MNIST images. In this benchmark task we plot results in fig. 2 for three different architectures and our best (LIVI 3) remains the same architecture as above. Here too, we nearly match the performance of deep ensembles on these two metrics. The other two architectures, LIVI 1 has 1764 dimensional noise input and one matrix multiplication hidden layer with 350 units while LIVI 2 has 900 dimensional noise input with 2 hidden matrix multiplication layers of 320 units each.

## 6.4 COMPARISON BETWEEN IMPLICIT VARIATIONAL APPROXIMATIONS

Variational inference for BNNs relies heavily on the expressivity of the family of approximations chosen to model the posterior. In our case the architecture of the generator represents the flexibility and overall modeling capability of the implicit variational density. We trained different architectures and noticed that generator architectures with more hidden layers perform better on in-distribution metrics like accuracy and log-likelihood. Additional hidden layers afford the generators the capacity to warp the input Gaussian noise into a suitable posterior distribution. On the other hand, the dimensionality of input noise becomes crucial for uncertainty quantification and OOD performance. We believe this is because the number of noise inputs are all the degrees of freedom available to the generator to model the parameters of the BNN. As such, the entropy of resulting posterior is

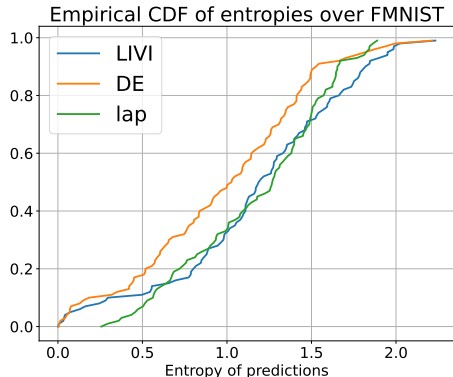

Figure 3: **Empirical CDF plot.** On the x-axis is the entropy of individual predictions and empirical CDF on the y-axis. As we count over the whole dataset, we would like to encounter more high entropy predictions closer to the value 2.30 on the x-axis.

directly dependent on the this factor. Although this number cannot be increased without repercussions because the base distribution and the number of samples affect the signal to noise ratio of our objective function eq. (29) and a very large $z$ results in large gradient variance hindering covergence, requiring more samples during training or higher number of iterations to converge. In these experiments we also noticed that down scaling only the prior log probability has a very positive effect on the results. This is due to the fact that the prior term regularises the generator, forcing it to find minimas close to itself, a standard normal distribution. The scale of this prior log probability term is significant in the ELBO and gradients of this term are detrimental to the overall optimisation process. Unlike cold-posteriors(Wenzel et al., 2020), we keep the gradients of the entropy term as is and only reduce regularisation by downscaling the prior.

As the last benchmark we opt to ascertain the quality of our model's posterior and the implied predictive uncertainties by plotting the empirical CDF of predictive entropies across OOD images(Lakshminarayanan et al., 2017; Louizos & Welling, 2017) in fig. 3. Given a model trained on MNIST, the predictions on a data point from a different distribution should be given a high entropy prediction like a uniform distribution. For this plot we first obtain entropies of the output softmax distributions for all the models across data points and use an empirical CDF to represent how many of these predictive entropies are closer to a uniform distribution which has an entropy of 2.3. Ideally, we are looking lines closer to the right bottom corner, i.e. the number of low-entropy or highly confident predictions should be less. We compare our model to MAP, deep ensembles and last-layer Laplace and find that our model trained on MNIST is quite competitive in the quality of uncertainty estimates for this test over FMNIST dataset. For this plot we use the best generator architecture with a LeNet5 BNN which is called LIVI 3 in the tests above.

# 7 CONCLUSION

In this paper we present a novel method for implicit variational inference for Bayesian Neural Networks that circumvents the need for a discriminator network to estimate intractable density ratios. We find that modelling the posterior with a highly flexible approximation indeed does have benefits. Our methods, in the wide range of variational approximations, get closer to the performance of deep ensembles, a non-probabilistic method on in distribution and out of distributions performance. Unlike conventional probabilistic methods we do not. One possible limitation of such hypernetworks can be generating massive parameter vectors for large neural networks. Works like Pawlowski et al. (2017); Shi et al. (2018) use different generator architectures to generate weights for each hidden layer in turn loose the information from modelling correlations across layers. Similarly this approach can be extended to use multiple smaller generators at the sacrifice of modelling correlations across layers.

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

# A   APPENDIX

## A.1   DETAILS ON APPROXIMATION OF THE DIFFERENTIAL ENTROPY

In this section we derive eq. (22). To simplify the derivation, we will use the notation $\boldsymbol{v}^2 = \boldsymbol{v}^\intercal \boldsymbol{v}$ for vectors and $M^2 = M^\intercal M$ for matrices.

Starting from the left hand side of eq. (22), we have that

$$\mathbb{E}_{\boldsymbol{z} \sim q(\boldsymbol{z})} \mathbb{E}_{\boldsymbol{\theta} \sim q(\boldsymbol{\theta}|\boldsymbol{z})}[g(\boldsymbol{\theta}, \boldsymbol{z})] = \mathbb{E}_{\boldsymbol{z} \sim q(\boldsymbol{z})} \mathbb{E}_{\boldsymbol{\theta} \sim q(\boldsymbol{\theta}|\boldsymbol{z})} \left[ \frac{1}{2}(\boldsymbol{\theta} - \boldsymbol{\mu}(\boldsymbol{z}))^\intercal C(\boldsymbol{z})^{-1}(\boldsymbol{\theta} - \boldsymbol{\mu}(\boldsymbol{z})) \right] \tag{34}$$

$$= \frac{1}{2}\mathbb{E}_{\boldsymbol{z} \sim q(\boldsymbol{z})} \mathbb{E}_{\boldsymbol{\theta} \sim q(\boldsymbol{\theta}|\boldsymbol{z})}[\text{tr}((\boldsymbol{\theta} - \boldsymbol{\mu}(\boldsymbol{z}))^\intercal C(\boldsymbol{z})^{-1}(\boldsymbol{\theta} - \boldsymbol{\mu}(\boldsymbol{z})))] \tag{35}$$

$$= \frac{1}{2}\mathbb{E}_{\boldsymbol{z} \sim q(\boldsymbol{z})}[\text{tr}(C(\boldsymbol{z})^{-1}\mathbb{E}_{\boldsymbol{\theta} \sim q(\boldsymbol{\theta}|\boldsymbol{z})}[(\boldsymbol{\theta} - \boldsymbol{\mu}(\boldsymbol{z}))^\intercal(\boldsymbol{\theta} - \boldsymbol{\mu}(\boldsymbol{z}))])] \tag{36}$$

The inner expectation simplifies to

$$\mathbb{E}_{\boldsymbol{\theta} \sim q(\boldsymbol{\theta}|\boldsymbol{z})}[(\boldsymbol{\theta} - \boldsymbol{\mu}(\boldsymbol{z}))^2] = \mathbb{E}_{\boldsymbol{\theta} \sim q(\boldsymbol{\theta}|\boldsymbol{z})}[(\boldsymbol{\theta} - g(\boldsymbol{z}) + \mathrm{D}g(\boldsymbol{z})\,\boldsymbol{z})^2] \tag{37}$$

$$= \mathbb{E}_{\boldsymbol{\theta} \sim q(\boldsymbol{\theta}|\boldsymbol{z})} \left[ (\boldsymbol{\theta} - g(\boldsymbol{z}))^2 + (\mathrm{D}g(\boldsymbol{z})\,\boldsymbol{z})^2 + 2(\boldsymbol{\theta} - g(\boldsymbol{z}))\,\mathrm{D}g(\boldsymbol{z})\,\boldsymbol{z} \right] \tag{38}$$

$$= \sigma^2 I_m + (\mathrm{D}g(\boldsymbol{z})\,\boldsymbol{z})^2, \tag{39}$$

where we that $\mathbb{E}_{\boldsymbol{\theta} \sim \mathcal{N}(\boldsymbol{\theta}|g(\boldsymbol{z}),\sigma^2 I_m)}[(\boldsymbol{\theta} - g(\boldsymbol{z}))] = 0$ and $\mathbb{E}_{\boldsymbol{\theta} \sim \mathcal{N}(\boldsymbol{\theta}|g(\boldsymbol{z}),\sigma^2 I_m)}[(\boldsymbol{\theta} - g(\boldsymbol{z}))^2] = \sigma^2 I_m$. If we plug in the result of eq. (39) into eq. (36), we obtain

$$\mathbb{E}_{\boldsymbol{z} \sim q(\boldsymbol{z})} \mathbb{E}_{\boldsymbol{\theta} \sim q(\boldsymbol{\theta}|\boldsymbol{z})}[g(\boldsymbol{\theta}, \boldsymbol{z})] = \frac{1}{2}\mathbb{E}_{\boldsymbol{z} \sim q(\boldsymbol{z})} \left[ \text{tr} \left( \left( \mathrm{D}g(\boldsymbol{z})^2 + \sigma^2 I_m \right)^{-1} \left( \sigma^2 I_m + (\mathrm{D}g(\boldsymbol{z})\,\boldsymbol{z})^2 \right) \right) \right]. \tag{40}$$

Note that eq. (40) could also be derived from eq. (34) using Petersen & Pedersen (2012, eq. 380) and some reordering the terms. Equations (37) to (39) also follows from Petersen & Pedersen (2012, eq. 325).

## A.2   EXPERIMENT DETAILS

We use the MMNN architecture as presented in Shi et al. (2018) for generating weights of the MLP BNN that was trained on MNIST as well as LeNet BNN that was used for all the OOD benchmarks. For the MLP experiment to compare with KIVI we used one MM network that generated all the parameters of the network Following architectures were tried for LeNet5 generators:

- Noise input - 25x25, 2 MM hidden layers with 250 units, output layer size 350x127.
- Noise input - 30x30, 2 MM hidden layers with 350 units, output layer same as above.
- Noise input - 35x35, 2 MM hidden layers with 350 units, output layer same as above.
- Noise input - 35x35, 3 MM hidden layers with 350 units, output layer same as above.
- Noise input - 38x38, 2 MM hidden layers with 350 units, output layer same as above.
- Noise input - 42x42, 2 MM hidden layers with 325 units, output layer same as above.

All the above architectures were trained without dataset augmentation and with a maximum of 6 samples per minibatch. The last architecture required higher number of samples due to gradient noise which is proportional to the dimensionality of input noise. This phenomenon has been widely observed in training high dimensional variational approximations (Osawa et al., 2019; Mohamed et al., 2020). As all the architectures are trained for 100 epochs with the same learning rate, increasing gradient noise can significantly deter convergence when the input noise dimensions are increased.

## A.3   PRIOR DOWNWEIGHTING

We choose to down scale the log prior probability in all the benchmark experiments. This term appears in the ELBO objective function and serves an important purpose. When the prior is chosen by

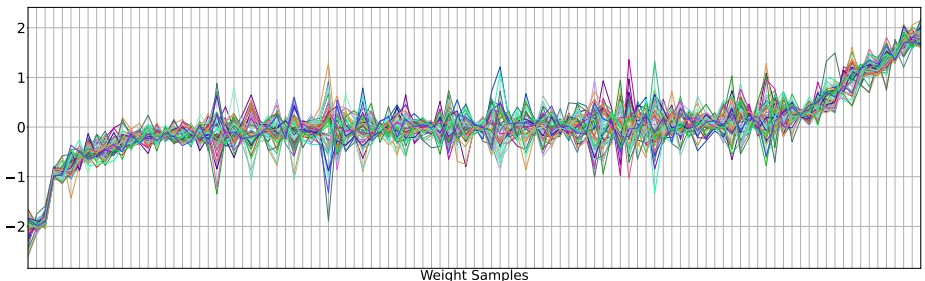

Figure 4: Zero-Centered weights

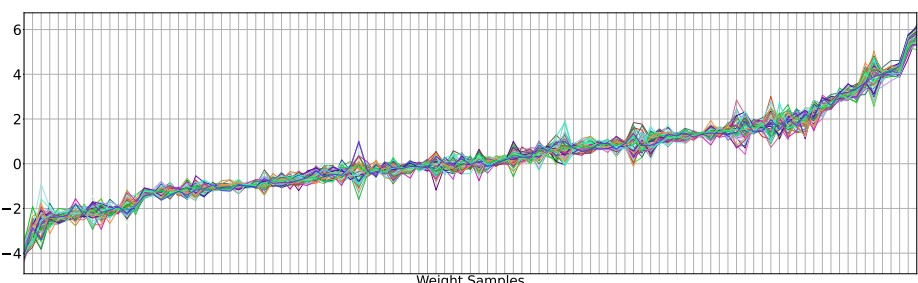

Figure 5: Weights activated after down-weighting the prior

domain experts this term ensures that the approximate posterior inferred is not too different from the intelligently chosen prior and hence the log prior probability of samples coming from the variational approximation should be high when the ELBO is being maximised. However, the choice of an appropriate prior is an active area of research in Bayesian deep learning(Fortuin, 2022) and this prior term and the regularisation effect is known to limit the variational approximation. $D_{\mathrm{KL}}[q(\theta)||p(\theta)]$ should be minimised as a result of the optimisation of ELBO using gradient ascent and when the prior is naively chosen to be a standard normal it forces most of the weights of the posterior to be zero-centered. This forces the model to look for minimas that are very close to 0 and has a detrimental effect on the in-distribution performance. We use the plotting tool used by Shi et al. (2018) to demonstrate this effect. The line-plot below has all of the weights of a neural network used to solve a toy regression task on the x-axis and their respective magnitudes on the y-axis. We chose to sort the weights in order of their magnitude as the positions of weights are not very informative in neural networks due to permutation invariance. In the first plot, most of the weights are zero-centred and are not very active, on the other hand the second plot shows what happens when we down weight the prior by just 0.1.

## A.4 DETAILS OF FIG. 1

In fig. 1 we compare both the objective functions presented in this work for training with implicit variational approximations to different methods for uncertainty quantification for neural networks. All models were trained for 10K iterations and had to learn observation noise present in the toy sinusoidal dataset. We deliberately removed a part of the data to see if the models tested were able to find in-between uncertainties. All methods were given the same sized 2 hidden layered MLP with 7 and 10 units respectively. We trained 5 networks with different seeds for Deep Ensembles and average their predictions to make the plot. The variance of the predictions were then used for the confidence bands in blue. We also train the model with an observation noise parameter. For MFVI, we used KL down weighting to get it to convergence and increase the weight in the end of training. For HMC we

Figure 6: Multimodal densities over weights using KDE

sample 5000 samples using the library `Pyro`. We also tried to make multiplicative normalizing flows converge for this dataset, but with even 20K parameters and training for 15K iteration with a very long learning rate did not help. We even tried KL down weighting to reduce the effect of the prior in the initial iterations but that did not work either.

## A.5   COMPUTATION GRAPH

Here we provide some details about how the combination of the joint generator-BNN model works. The Bayesian neural network classes for all types of architectures(feed forward, convolutions, etc.) require a generator in the `init` function. As such, the generator networks reside inside the BNN and reparametrise it with a simple `sample_parameters` function. The most important part of this kind of implementation was the layers themselves. **PyTorch** provides different kinds of mutable layer implementations in `nn.module` but these layers do not expose their state i.e. their parameters in a manner that allows changes on the fly during training. We reimplemented the layers allowing such resampling to occur with the generator. In the `init` function of the BNN, we generate one set of parameters with the generator, package it in a `dict` that has the weight sample as well as a index to know the number of weights used by a previous layer. This counter index is updated by each layer in their `init` and `sample_parameters` function. As such, only the parameters of the generator are trainable, the parameters of the BNN are switchable and relay gradients to the generator via the likelihood or the entropy term.

## A.6   KDE PLOT

Figure 6 shows a KDE plot of weights randomly chosen from samples obtained from a trained generator.

