# OpenReview forum: "Linearised Implicit Variational Inference"
_ICLR.cc/2023/Conference — Submitted to ICLR 2023_

### Official Review · Reviewer_xXZT · 2022-10-23

**Confidence:** 4
**Correctness:** 2
**Technical Novelty And Significance:** 3
**Empirical Novelty And Significance:** 2
**Recommendation:** 3

**Clarity, Quality, Novelty And Reproducibility:**

**Clarity** The methodological section is clear. The experimental section is a bit underwhelming in this regard, e.g. there are different variations of the proposed method that are not defined at all ('acc-jac' and 'diff-lb' in Fig 1) or barely motivated (LIVI 1-3). I would suggest the authors state explicitly what questions they are trying to address with these variant.
**Quality** While the methodological derivations are correct and interesting, the experimental evaluation is severely lacking.
**Novelty* The approach is novel.

**Strength And Weaknesses:**

Strengths:
* Generating parameters by a neural network is an appealing direction for variational as it allows for potentially complex, multi-modal variational distributions. This paper proposes a coherent and principled approach for this.
* The technical exposition is extremely clear.

Weaknesses:
* The experimental evaluation is extremely small scale, not going beyond MNIST, which is not exactly informative.
* There are unfortunately no ablation studies or relevant qualitative experiments. Given that the paper makes multiple approximations, I really would have wanted to see some more in-depth analysis of these choices rather than fairly tangential experiments on the architecture of the generator (I know this matter, but does not seem exactly relevant for the main text).
* There is a lengthy discussion around down-weighing the prior term in the ELBO. I know it is common practice to temper the KL divergence in the variational objective and have used such tricks myself, however I find this somewhat contradicts the motivation of the paper to approximate the entropy term in the objective. Why bother approximating the entropy if we're not using the ELBO in a principled way anyway?

Minor comments:
* I don't understand why the variance on the parameters is taken to the limit of 0 rather than being treated as a variational parameter.
* I'm not entirely sure whether this equivalence holds, but taking the outer product of the jacobians that the paper uses as equal to the Fisher, I wonder whether instead of using an iterative approach for calculating the singular value the structure of the Fisher could be exploited as e.g. in (Ritter et al., A scalable Laplace approximation for neural networks, ICLR 2018) for efficiently calculating eigenvalues/log determinant.
* I would suggest adding HMC results to the UCI benchmark for reference.
* FIg 3 should include reference values for the in-distribution test data, OOD entropies are meaningless in isolation.

**Summary Of The Paper:**

The paper proposes a variational inference method based on sampling model parameters from a generator neural network. It calculates the entropy of resulting implicit variational distribution by linearizing the network, resulting in a Gaussian entropy. Further, the paper proposes to bound the costly log determinant calculation of the variance by the maximum eigenvalue (via the maximum singular value of the Jacobian). It reports competitive performance on UCI regresson and MNIST uncertainty estimation benchmarks with deep ensembles, a kernel-based implicit VI method and a last-layer Laplace approximation.

**Summary Of The Review:**

While in principle I like the approach that the paper takes -- I find it much more coherent and principled than prior works on implicit VI -- the evaluation is just not sufficient to recommend acceptance. If the authors want to take a primarily quantitative route, I'm afraid that some larger scale experiments, at least some ResNets on CIFAR, would be necessary to be convincing for the community. MNIST is simply not a meaningful quantitative benchmark anymore.

Personally, I would find some more qualitiative analyses on the approximations that the paper makes insightful. It's great that there are e.g. some histograms of samples that show the generator is multi-modal, but it is not too surprising that a neural network is capable of doing that. The main question is whether the proposed approach approximates the entropy decently well and what happens when the prior term is not down-weighted, so that we can attribute resulting performance gain to the method being an approximately Bayesian one rather than an arbitrary stochastic neural network.

---

### Official Review · Reviewer_mGTw · 2022-10-24

**Confidence:** 4
**Correctness:** 3
**Technical Novelty And Significance:** 2
**Empirical Novelty And Significance:** 2
**Recommendation:** 3

**Clarity, Quality, Novelty And Reproducibility:**

Given the above reviews, the clarity and reproducibility are good but the novelty and quality are poor.

**Strength And Weaknesses:**

# Strength
- The paper develops a new differentiation bound on the entropy of implicit distributions. This may be useful for the probabilistic inference community.
- The literature review is relatively thorough.

# Weaknesses
## Major issues
- Regarding the ill-defined KL when d < m, the authors should discuss the quasi-KL measure [1] to make the paper more convincing.
- My major concern about this paper is that there are too many approximations such that I am not convinced of the fidelity and reliability of the yielded bound. One approximation is about local linearization around the sample z. The approx. error here can be arbitrarily large? The second approximation lies in eq 23. You really use a $\simga^2$ that approaches 0? But if doing so, you fall back to the ill-defined KL... The third approximation is eq 31, where Jacobians are replaced with their singular values. The approximation error here can be bounded but currently, the overall approximation error is unmeasurable. Thus, I question the reliability of this method and believe more theoretical analysis regarding the tightness of you bound of entropy is required.
- The main technical novelty lies in the local linearization of the generator, which in my opinion, is limited. As said, more discussion or analyses on local linearization are needed.
- The biggest limitation of this method is its poor scalability. It is two-fold. (1) The generator cannot trivially generate millions of parameters for modern NNs as it cannot have that wide output layer. (2) The singular values of Jacobian are expensive to estimate; even the Jacobians themselves cannot be easily estimated for modern NNs. As a result, the method cannot be applied to realistic datasets and models. Results on at least cifar-10 are appreciated.
- Why the closely related KIVI is not included in the MNIST exps?

## Minor issues
- The writing is not good enough and there are typos. An example is the first paragraph of sec 3.1.
- By inspecting figure1, I don't think LIVI is as good as HMC, DE, and even MNF. Though the authors highlight LIVI captures in-between uncertainty, but it seems that it is not good enough and at least worse than that of even MNF. Can the authors provide a quantitative estimation of the quality of the predictive distributions of these methods using something like the divergence from the ground-truth predictive distribution (provided by HMC in my opinion)? By the way, why isn't the closely related KIVI included here?

[1] Variational Bayesian dropout: pitfalls and fixes

**Summary Of The Paper:**

This paper posits a new variational inference method based on implicit variational distributions. It develops a bound for estimating the entropy of the implicit distribution involved in the ELBO based on local linearization. The authors then use a differentiable numerical lower bound on the Jacobians of the generator to mitigate computational concerns. Experiments are conducted on UCI regression and MNIST classification.

**Summary Of The Review:**

Given the issues of approximations and limitations, I vote for rejecting this paper.

---

### Official Review · Reviewer_9D7T · 2022-10-24

**Confidence:** 4
**Correctness:** 3
**Technical Novelty And Significance:** 2
**Empirical Novelty And Significance:** 3
**Recommendation:** 5

**Clarity, Quality, Novelty And Reproducibility:**

# Clarity

* The paper is mostly self-contained, with clear explanations about the different concepts needed to understand the contribution. The derivation of the simplified version of the entropy term is detailed and done step-by-step, which helps understanding the procedure.

* The definition of the prior as an implicit distribution is implied, but never explicitly shoen, which would be clearer. Please, provide an explicit expression in Sections 2 or 3.

* Some typos:
  * "constant constant" (paragraph above Eq.11)
  * The last two lines of the paragraph above Eq.11 do not make much sense
  * Check first sentence of section 3.1
  * Eq.(13) and the following text seem to be referring to different things. Please, check carefully this discussion.
  * Is the covariance term of Eq.(16) first's step right?

# Quality

I think it can be an interesting contribution to the community interested in implicit-distribution-based inference. I wish the experimental support was a bit sturdier, since the whole idea here is to provide better uncertainty estimates than other methods.

# Novelty

The method itself does not seem very novel, rather a combination of previous ideas but for the derivation of the new objective function.

# Reproducibility

The authors do not mention whether they will provide the code for the method or not, and therefore it remains to be seen if it is easily reproducible.


**Strength And Weaknesses:**

# Strengths:

* The constructed model is reasonable and simplifies terms of the objective function that are usually very hard to deal with.

* The final objective function obtained seems an interesting step for implicit-distribution-based methods.

* The paper is clearly written for the most part and can be followed easily.

# Weaknesses:

* The proposed system does not seem too different from previous proposals. As the authors mention, this is highly related to Bayesian hypernetworks and normalizing flows, and somewhat could be seen as a particular combination of both concepts.

* Since the motivation behind the contribution is related to providing better uncertainty estimates for BNNs, I think the authors should provide a stronger experimental phase on which this is shown more extensively (e.g. adding comparisons against HMC in toy datasets and comparing with other methods that have shown high performance in this regard, s.a. [2]).

* In several points of the article where previous literature on the topic is covered, I cannot help but notice that some important contributions are missing. For instance, [1] should be clearly mentioned here since it is highly related to the topic, and this applies both to the initial setup on page 2 as well as to the Related Work section. Moreover, both [1] and [2] could (and maybe should) be considered as benchmarks to compare against. Moreover, there has been an extensive ongoing research on the implicit approach applied to the function-space formulation of BNNs which is never mentioned. These methods have shown improved performance and several relevant properties that the regular weight-space formulation fails to reproduce. Some relevant examples here are [3,4,5,6], among others. In particular, these methods extend the formulation of Eq.(6) to implicit stochastic processes.

* Scalability studies and a detailed comparison with other methods is not included anywhere. I strongly suggest the authors to provide some insights here.

## References:

[1] Mescheder, Lars, Sebastian Nowozin, and Andreas Geiger. "Adversarial variational bayes: Unifying variational autoencoders and generative adversarial networks." International Conference on Machine Learning. PMLR, 2017.

[2] Santana, S. R., & Hernández-Lobato, D. (2022). Adversarial α-divergence minimization for Bayesian approximate inference. Neurocomputing, 471, 260-274.

[3] Ma, C., Li, Y., and Hernández-Lobato, J. M. (2019). “Variational implicit processes”. In: International Conference on Machine Learning, pp. 4222–4233.

[4] Sun, S., Zhang, G., Shi, J., and Grosse, R. (2019). “Functional variational Bayesian neural networks”. In: International Conference on Learning Representations.

[5] Ma, C., & Hernández-Lobato, J. M. (2021). Functional variational inference based on stochastic process generators. Advances in Neural Information Processing Systems, 34, 21795-21807.

[6] Rodrı́guez-Santana, S., Zaldivar, B., & Hernandez-Lobato, D. (2022, June). Function-space Inference with Sparse Implicit Processes. In International Conference on Machine Learning (pp. 18723-18740). PMLR.

**Summary Of The Paper:**

Exact posteriors over parameters in BNNs are usually more complex than the prescribed-density approximations that are usually imposed in VI. The usage of implicit variational distributions allows for an increase in the flexibility of these methods. In this paper, the authors propose a novel bound for training BNNs in such schemes. This is achieved through the combination of a Gaussian deep-latent variable model and a Laplace approximation of the generator of the implicit distribution samples, which simplifies the estimation of the KL regularization term in the original ELBO objective function.

**Summary Of The Review:**

The method seems interesting, and although strongly based on previous ideas, could prove to be an important contribution to the research community. The authors should consider comparing against some other state-of-the-art approaches and provide stronger experimental evidence for the properties of the proposed method. However, the proposal is promising and could be of use to other researchers.

---

### Author Response · Authors · 2022-11-20
**Thank you for an insightful and thorough review**

We thank all the reviewers for their insightful review of our work and valuable feedback. In particular, we appreciate that they found the approach to be “novel” (Reviewer xXZT), “much more coherent and principled than prior works on implicit VI” (Reviewer xXZT), “reasonable and simplifies terms of the objective function that are usually very hard to deal with” (Reviewer 9D7T), and “useful for the probabilistic inference community” (Reviewer mGTw).

We agree with the reviewers that the experiment could be more extensive, and we will consider the reviewers’ feedback for the next iteration of the paper. In particular, we agree with Reviewer xXZT that the quality of the entropy approximation needs deeper analysis.
We thank Reviewer 97DT for pointing out the reference [2]; we were unaware of this and will include it in the next iteration of our paper. We agree with Reviewer 97DT that references [3,4,5,6] are relevant to BNN literature and quite impressive in their own right, but our work focuses on inferring flexible posteriors over neural network weights and hence is quite different in formulation to any of these function space inference works. In our opinion, while these works could be discussed generally in the paper, their loose relevance to weight space inference does not warrant an extensive discussion in a conference submission. We also thank
Reviewer mGTw for the reference [1], we were unaware of this work and will indeed include the quasi-divergence measure in the next iteration of our paper.

To Reviewer xXZT, we would like to emphasise that prior downweighing and KL tempering are quite different and the latter tampers with the gradients of the entropy term reducing its influence. This term and its approximation are central to our work and its maximization leads to a more diffused and possibly multimodal variational density which is our aim. With the downweighing of the prior we only intended to reduce the unwanted regularisation effect from a naively chosen prior which is meaningless in the function space and is completely uncorrelated in weights. This also gave us a significant performance boost(correlation plot in the appendix). This prior’s ill effects have been noticed by other works[7][8] and the right choice of prior is by itself a challenging problem.

We strongly disagree with any comments on the lack of technical novelty (Reviewer 9D7T and mGTw) and expect stronger arguments supporting the reviewers’ stand. We will clarify the novelty in the next iteration of the paper.
We thank the reviewers for pointing out typos and errors. We will address these in the next iteration of the paper.
1. Variational Bayesian dropout: pitfalls and fixes
2. Santana, S. R., & Hernández-Lobato, D. (2022). Adversarial α-divergence minimization for Bayesian approximate inference. Neurocomputing, 471, 260-274.
3. Ma, C., Li, Y., and Hernández-Lobato, J. M. (2019). “Variational implicit processes”. In: International Conference on Machine Learning, pp. 4222–4233.
4. Sun, S., Zhang, G., Shi, J., and Grosse, R. (2019). “Functional variational Bayesian neural networks”. In: International Conference on Learning Representations.
5. Ma, C., & Hernández-Lobato, J. M. (2021). Functional variational inference based on stochastic process generators. Advances in Neural Information Processing Systems, 34, 21795-21807.
 6. Rodrı́guez-Santana, S., Zaldivar, B., & Hernandez-Lobato, D. (2022, June). Function-space Inference with Sparse Implicit Processes. In International Conference on Machine Learning (pp. 18723-18740). PMLR.
7. Bayesian Neural Network Priors Revisited, Fortuin et. al. 2022
8. Wide Mean-Field Bayesian Neural Networks Ignore the Data, Coker et. al. 2022

---

### Decision · Program_Chairs · 2023-01-20

**Decision:**

Reject

**Justification For Why Not Higher Score:**

Direction is good, but analysis is insufficient for acceptance at this moment.

**Justification For Why Not Lower Score:**

N/A

**Metareview: Summary, Strengths And Weaknesses:**

Interesting new variational inference approach. Approximation of the entropy could use a better analysis. Experiments have been considered insufficient. AC fully appreciated the authors' summary posted after the reviews and hopes that the reviewers have appreciated it too.